# Learning to Prove Theorems by Learning to Generate Theorems

**Mingzhe Wang**
Princeton University
mingzhew@cs.princeton.edu

**Jia Deng**
Princeton University
jiadeng@cs.princeton.edu

## Abstract

We consider the task of automated theorem proving, a key AI task. Deep learning has shown promise for training theorem provers, but there are limited human-written theorems and proofs available for supervised learning. To address this limitation, we propose to learn a neural generator that automatically synthesizes theorems and proofs for the purpose of training a theorem prover. Experiments on real-world tasks demonstrate that synthetic data from our approach improves the theorem prover and advances the state of the art of automated theorem proving in Metamath. Code is available at https://github.com/princeton-vl/MetaGen.

## 1   Introduction

Automated theorem proving aims to automatically generate a proof given a conjecture (the target theorem) and a knowledge base of known facts, all expressed in a formal language. Automated theorem proving is useful in a wide range of applications, including the verification and synthesis of software and hardware systems (Gu et al., 2016; Darvas et al., 2005; Kern & Greenstreet, 1999).

Automated theorem proving boils down to a search problem: finding the sequence of symbol manipulations that generate a valid proof. The fundamental challenge lies in the explosion of search space, in particular with long proofs and large knowledge bases. The success of theorem proving thus relies on effective heuristics that guide the prover by deciding the next step the prover should take.

Deep learning has emerged as a promising approach to learning search heuristics in an automated theorem prover (Irving et al., 2016; Whalen, 2016; Loos et al., 2017; Bansal et al., 2019a; Lee et al., 2019). The search process fundamentally reduces to a sequence of actions on manipulating a set of symbols. Thus a deep network can be trained to select the best action at each step.

A key challenge is how to train such networks. Prior work has used human-written theorems and proofs to perform imitation learning and has shown promising results (Loos et al., 2017; Yang & Deng, 2019; Whalen, 2016; Paliwal et al., 2019). The training data consists of theorems and proofs manually written by human experts in a formal language, and the prover is trained to imitate the proof steps demonstrated by humans.

However, relying on human-written data has a major drawback: such data has limited availability and scalability. Writing theorems and proofs in a formal language requires highly specialized knowledge and skills, including mathematics, computer programming, and proficiency in the particular formal language. For a CS graduate student, it can take months to master a new formal language such as Mizar, Metamath or HOLight (Wiedijk, 2003), after which it can take days to formalize a single page of a math textbook. This makes it impractical to crowdsource human-written proofs at large scale.

In this paper, we propose to train a theorem prover using synthetic data. The basic idea is to construct a *generator* that automatically synthesizes new theorems and their proofs, which serve to augment human-written data for training the prover.

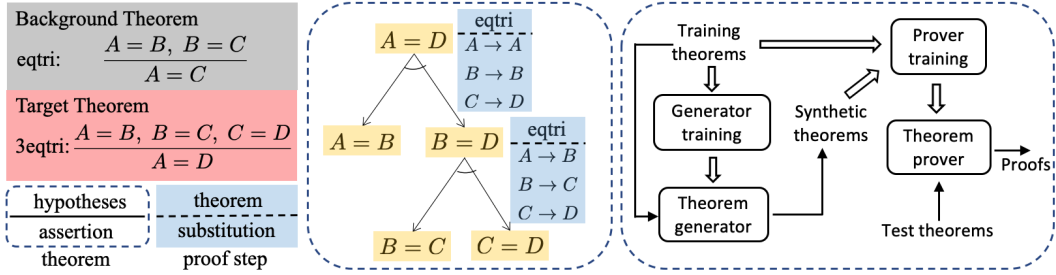

Figure 1: Left: A proof task. Middle: The proof tree of the theorem 3eqtri. Each leaf node is a hypothesis and each internal node corresponds to a proof step. Right: The overview of our approach.

To generate a new theorem and its proof, the generator performs a sequence of symbol manipulations, similar to a prover. It repeatedly applies inference rules on a set of existing theorems and combines their proofs to form the proof of the new theorem. It is important to note that despite the similarity of operations, the generator has a much easier task than the prover. The generator just needs to generate *some* new theorem of *its own choice*, whereas the prover needs to find the proof for a particular target theorem specified by someone else.

One challenge of generating synthetic theorems is that there are infinitely many possibilities but the prover can only use a finite amount of them during training. Not all theorems are equally useful as training data. Thus a key question is how to generate synthetic theorems that are more useful. To this end we make the generator learnable by parameterizing it with deep networks.

We hypothesize that the generated data will be more useful if they are similar to human-written data. Therefore we use human-written data to train a generator. We consider two scenarios. If the human-written data consist of both theorem statements and their proofs, we train the generator to follow the proof steps in the forward direction, so that a well-trained generator would derive theorems humans tend to derive. If the human-written data consist of only theorem statements but not their proofs, i.e. no human actions to imitate, we use reinforcement learning to let the generator discover good actions that lead to synthetic theorems that are similar to the human-written theorems. To measure similarity between synthetic theorems and human theorems, we use a discriminator trained to distinguish the human theorems from synthetic ones, similar to GANs (Goodfellow et al., 2014).

We instantiate our approach in Metamath (Megill & Wheeler, 2019), a popular language for formal mathematics, and with Holophrasm (Whalen, 2016), a Metamath neural prover. We propose a neural theorem generator called "MetaGen", which synthesizes new theorems and their proofs expressed in the formalism of Metamath. To the best of our knowledge, MetaGen is the first neural generator of synthetic training data for theorem proving. Experiments on real-world Metamath tasks show that synthetic data from MetaGen can help train better provers, advancing the state of art in theorem proving on Metamath.

## 2 Related Work

**Automated theorem proving** Our work is related to prior work on learning to prove theorems (Whalen, 2016; Gauthier et al., 2018; Bansal et al., 2019a; Yang & Deng, 2019; Loos et al., 2017; Balunovic et al., 2018; Kaliszyk et al., 2018; Bansal et al., 2019b; Polu & Sutskever, 2020). Our work directly builds off of Holophrasm (Whalen, 2016), a neural-augmented theorem prover for Metamath. It contains three deep networks to generate actions and initial values to guide proof search following the UCT algorithm (Kocsis & Szepesvári, 2006). Polu & Sutskever (2020) also build the theorem prover for Metamath by adopting the GPT-like network architectures and pretraining methods and generating proof steps autoregressively.

TacticToe (Gauthier et al., 2018), DeepHOL (Bansal et al., 2019a) and ASTactic (Yang & Deng, 2019) are learning-based theorem provers based on interactive theorem provers HOL4 (Slind & Norrish, 2008), HOL Light (Harrison, 2009) and Coq (Bertot & Castéran, 2004) respectively. Paliwal et al. (2019) improves DeepHOL by representing formulas as graphs. Loos et al. (2017) proposes to learn clause selection by deep learning inside the first-order logic prover E (Schulz, 2002).

All of these methods are orthogonal to our approach because all of their provers are learned from human-written training data, whereas our prover is trained from human data augmented with synthetic data. Our contribution is on the generation of synthetic data and using such data to train a prover.

Kaliszyk et al. (2018); Bansal et al. (2019a,b); Balunovic et al. (2018) use reinforcement learning to train provers with only human-written theorems or SMT conjectures but not proofs. During training, a prover collects rewards only upon finding full proofs. In contrast, we always train our prover using imitation learning. Under the same setting with only human-written theorems but not proofs, we use reinforcement learning to train our generator, whose reward is the similarity between a generated theorem and human-written theorems, as measured by an adversarial discriminator. Our reinforcement learning task is much easier because the reward is continuous and there are many ways to generate theorems similar to human-written ones.

**Synthetic theorem generation** Zombori et al. (2019); Fawzi et al. (2019) construct theorem provers by training on randomly generated synthetic theorems and evaluate the learned prover on synthetic theorems. The main difference of our approach is that our generator is optimized through learning, as opposed to random generation.

Kaliszyk et al. (2018); Jakubův & Urban (2019); Urban et al. (2008); Kaliszyk et al. (2014); Piotrowski & Urban (2018) train theorem provers iteratively. They repeatedly apply the trained prover on existing human theorems and generate new machine proofs to train the prover further. In these methods, only new proofs are synthesized and the synthetic proofs are only for existing human theorems; no new theorems are synthesized. In contrast, our approach synthesizes both new theorems and new proofs which could cover a much larger space of possible derivations than the proofs of existing human theorems.

Urban (2004); Kaliszyk & Urban (2015); Kaliszyk et al. (2015) extract proof tasks from the proofs of human-written theorems, such as the intermediate inference steps or their variants. That is, they extract "sub-proofs" from existing proofs. In contrast, we generate entirely new theorems and new proofs that are not part of any existing proofs.

Our work is also related to the line of work on conjecturing (Chvalovský et al., 2019; Urban & Jakubův, 2020; Colton, 2012), which aims to generate mathematical conjectures automatically. The generated conjectures are not necessarily true, and their proofs are not required. In contrast, each of our synthetic theorem is guaranteed to be correct and its proof is automatically available.

**Automatic goal generation by self-play** Our work is similar to the line of work in reinforcement learning (Florensa et al., 2018; Sukhbaatar et al., 2017, 2018; Durugkar & Stone, 2018) that deploys two agents in adversary self-play, where one agent to generate tasks for another agent to accomplish. We pursue similar ideas in the new context of theorem proving by learning to generate synthetic theorems to train the prover. Also of note is that we have no adversarial self-play. The goal of the generator is to discover novel theorems similar to human-written ones, not to beat the prover.

Recently, Huang (2019) introduced a two-player game which encourages players to learn to predict the consistency of logical formulas by self-play. These two players behave symmetrically and compete with each other in the game. In contrast, our generator and prover execute different tasks, and are co-operative. In addition, their game remains a theoretical proposal without any empirical validation, whereas we have performed experiments on large-scale data.

# 3 Background on Metamath

Metamath is a language for developing formal mathematics. It is one of the simplest formal systems. It has only one inference rule, called *substitution*, but is universally applicable in formalizing a large portion of mathematics [1] and different types of logic (Megill & Wheeler, 2019).

**Expression and theorem** A basic building block of Metamath is expressions. An expression is a sequence of tokens that follows a set of grammar rules called "generating axioms". A token is either a constant or a variable. For example, $x + 2 * y = y + (x + y)$ is an expression, where $x$ and $y$ are two variables. Each expression corresponds to a unique parse tree where each internal node represents a generating axiom and each leaf node is a token.

A theorem consists of a set of expressions, one expression as its *assertion* and zero or more expressions as its *hypotheses*. The theorem can be understood to state that the hypotheses (e.g. $x^2 = 1$ and $x > 0$) entail the assertion (e.g. $x = 1$). Some examples of theorems are shown in Figure 1.

**Substitution** The only inference rule in Metamath is *substitution*, which transforms one expression by replacing each variable with a non-empty new expression. For example, the expression $A = B$ can be transformed to $E + F = C * D$ by the substitution $A \rightarrow E + F$ and $B \rightarrow C * D$.

Given two expressions $a$ and $b$, we say $b$ can *reach* $a$ or $a$ is *reachable* from $b$ if there exists a substitution that transforms $b$ to $a$. This is equivalent to saying that the parse tree of $b$ can be obtained by "trimming" the parse tree of $a$—repeatedly picking an internal node, removing all its descendants, and replacing it with a variable node. Reachability can be checked by comparing parse trees; an algorithm is described in the supplementary material.

**Proof step** A *proof step* is the basic unit of reasoning. A proof step in Metamath has two parts: (1) a theorem and (2) a *substitution* that maps each variable in the theorem to a new expression. A proof step serves to establish entailment between expressions based on the invoked theorem. For example, let $t$ be the theorem over1i, with the hypothesis $A = B$ and the assertion $(A\ F\ C) = (B\ F\ C)$, where $\{A, B, C, F\}$ is the set of variables in $t$. Let $\phi$ be a substitution that maps each variable in $t$ to a new expression: $A \rightarrow 2$, $B \rightarrow (1+1)$, $C \rightarrow 2$ and $F \rightarrow +$. By replacing variables in $t$ with their corresponding expressions given by $\phi$, we have a new hypothesis $2 = (1+1)$ and a new assertion $(2+2) = ((1+1)+2)$ This proof step $(t, \phi)$ establishes that the new hypothesis $2 = (1+1)$ entails the new assertion $(2+2) = ((1+1)+2)$ based on theorem $t$. The new assertion is called the conclusion and the new hypothesis is called the precondition. Because a theorem has one assertion and zero or more hypotheses, a proof step thus has one conclusion and zero or more preconditions.

**Proof** A theorem is proved if we can construct a *proof tree* that connects the hypotheses of the theorem to its assertion through entailment. The root node of a proof tree is the assertion of the theorem. Each leaf node of the tree is either a hypothesis of the theorem or empty. Each internal node of the tree is an expression and is associated with a proof step that uses an pre-existing theorem, together with an appropriate substitution, to establish the entailment of this internal node by its child nodes. Note that if an internal node has an empty child, it means that the proof step has no preconditions. An example proof tree is shown in Figure 1.

A *proof* is a sequence of proof steps that can be obtained by traversing a proof tree in pre-order. This linearized proof is an equivalent to the tree representation. In this work we will use "proof" and "proof tree" interchangeably.

**Corpus** A *corpus* consists of a set of axioms and a sequence of theorems and their corresponding proofs. The proof of each theorem uses only the axioms and the preceding theorems.

# 4 Approach

**Task setup** We use the standard theorem proving setup in prior work Irving et al. (2016); Bansal et al. (2019a); Whalen (2016). A *proof task* consists of a *target theorem* (or "target" in short) to be proved and a set of *background theorems* to be used as known facts. For each theorem in a corpus, we construct a proof task using the theorem as the target theorem and all preceding theorems (i.e. the theorems that humans had available when they were proving the target theorem) as the background theorems. In other words, each theorem in the corpus corresponds to a unique proof task that uses the theorem as the target. We randomly split all theorems into three disjoint sets: a training set, a validation set, and a test set. Accordingly, we have three corresponding sets of proof tasks using the theorems as targets. More details about this setup in the supplemental material.

## 4.1 Generator

We propose *MetaGen*, a neural generator that performs forward reasoning to synthesize theorems. It takes a set of training proof tasks as input and outputs a set of synthetic theorems. These synthetic theorems are then combined with original training proof tasks to train the theorem prover (as shown in the right of Fig. 1). The basic operation is generating a proof step—selecting an existing theorem and constructing a substitution. From this single proof step we can derive a new theorem. Now, we can treat this new theorem as an existing theorem and repeat to generate additional new theorems.

One issue requiring special handling is avoiding generating "meaningless" theorems. A meaningless theorem is one that includes falsehood in its hypotheses—as a result it is always provable regardless what the assertion says. It is possible to generate such a theorem if we allow arbitrary substitutions in constructing a proof step. For example, the hypothesis $A = B$ can be substituted into $1 = 2$. Such theorems are valid but unlikely to be useful as training data.

To avoid meaningless theorems, in constructing a proof step, we require that each new hypothesis produced by substitution must be identical to the full expression of a node in an existing proof tree (either the root, a leaf, or an internal node), such as the five expressions in yellow boxes in Fig. 1. This prevents introducing false expressions as hypotheses, provided that the existing proofs have no false expressions. See supplemental material about more discussion on meaningless theorems

A second issue is generating new theorems with multi-step proofs. A single proof step gives a shallow tree. To generate theorems with longer proofs, we "graft" this shallow tree with existing proof trees or subtrees. For a leaf node $e$ of the shallow tree, we can replace it with an existing proof tree (or subtree) whose root node is also $e$. For example, suppose the shallow tree proves that $x^2 = 1$ and $x > 0$ entail $x = 1$, and there already exists another tree proving that $x^3 > 0$ entails $x > 0$. Then we can join the two trees to generate a new tree proving that $x^3 > 0$ and $x^2 = 1$ entail $x = 1$.

To generate theorems and proofs more similar to human-written ones, we impose an additional constraint that a synthesized proof step can only invoke a theorem that has appeared as a background theorem in a training proof task. This is because in the ground-truth proof for a proof task, only the background theorems are invoked in proof steps. This means that we do not invoke any synthesized theorems. To implement this constraint, the generator constructs proof steps using a restricted set of "invocable" theorems pre-specified as input to the generator.

**Initializing existing proof trees** The generator takes as input a set $E$ of existing theorems and optionally their proof trees, and a set $I$ of invocable theorems, where $E$ and $I$ are the union of the target and background theorems of the training proof tasks respectively. To enable tree grafting, it first builds a set $G$ of existing proof trees. For every theorem in $E$, if its proof tree is available, for every node $e$ in its proof tree, we add to $G$ the subtree that is rooted at $e$ and contains all nodes below $e$. Otherwise, we add to $G$ every hypothesis of this theorem as a single node proof tree.

Two proof trees are considered equivalent if they have the same root node and the same leaf nodes, i.e. they prove the same theorem. Among equivalent trees, we only keep the smallest one. As a result, $G$ contains all sub-proof trees from all the existing theorems that can be grafted to a new proof step.

**Generating new theorems** To generate a new theorem, the key procedure is to construct a proof step and a set $S$ of existing proof trees such that $S$ is a subset of $G$ and each precondition of this proof step matches the root node of a proof tree in $S$. This is achieved in three steps as follows:

1. Pick an invocable theorem $t \in I$ according to the frequencies of invocable theorems being used in the proofs of the existing theorems.

2. Initialize the set $S$ of proof trees as empty. Initialize the substitution $\phi$ for $t$ as empty. For each hypothesis $h$ of theorem $t$, apply the current substitution $\phi$ to hypothesis $h$ to obtain the transformed expression $h(\phi)$, find all *compatible* proof trees, those whose root nodes are reachable from $h(\phi)$—$h(\phi)$ can be transformed to the root nodes by substitution, which can be determined by comparing parse trees—and perform the following:

   - Select a compatible proof tree $c$ using a *relevance network* (to be described later). For each variable that has not been substituted in $h$, update $\phi$ by assigning the variable a substitute expression to match the root of $c$. Add tree $c$ to set $S$.

   If no compatible proof tree exists, go to Step 1 and rebuild this proof step from scratch.

3. If a variable appears in a hypothesis of $t$, its substitution has been determined by matching this hypothesis with the root of a compatible proof tree. For the remaining variables that appear exclusively in the assertion of $t$, use a *subtitution network* (to be described later) to generate substitute expressions for them.

This proof step gives a one-step proof tree, which we expand to a multi-step proof tree by grafting the trees in set $S$ onto its leaves. This multi-step proof tree is added to $G$ for subsequent generation. We repeat this procedure to get a set of synthetic theorems (pseudo-code in the supplementary material).

**Relevance network of generator** The *relevance network* in step 2 is a deep network trained to pick a proof tree from a set of candidates by scoring and ranking them. It uses the same design as the relevance network in Holophrasm Whalen (2016) (see Sec. 4.2) but has different inputs and purposes. It takes two sequences of tokens as input. One input sequence represents the root and leaf nodes of a proof tree. The other sequence consists of two parts. One part represents the leaf nodes of the proof trees that have been selected for preceding hypotheses (the hypotheses are processed one by one). The other part represents the assertion and hypotheses of the invocable theorem transformed by the current substitution, except for the current hypothesis to be processed which is represented by a special token. Two GRU encoders convert each input sequence to an embedding vector, followed by a bilinear layer to output a score from the two vectors. In practice, we limit the number of candidate trees to 2000 for tractability.

**Substitution network of generator** The substitution network generates the substitution for a target variable of an invocable theorem. It uses the same design as the "generation network" in Holophrasm Whalen (2016) (see Sec. 4.2) but has different inputs and purposes. It is a sequence-to-sequence model with the encoder-decoder GRU network. It takes as input the sequence of tokens that represents the assertion of the invocable theorem and the leaf nodes of the existing proof trees that have been selected to construct a proof step. The target variable is represented by a special token. The network outputs a sequence of tokens, sampled one by one based on the softmax probabilities.

**Generator training** We propose two strategies to train the relevance network and the substitution network, depending on the availability of human-written proofs.

Our generator can work without learnable parameters if we remove the two deep network and sample new proof steps by randomly picking existing proof trees and generating substitutions. We call such a generator as *MetaGen-Rand*.

Given human-written proofs, we train *MetaGen-IL* by imitation learning. Given a proof step $(t, \phi)$ in a human-written proof tree $\mathbf{s}$, each transformed hypothesis $h(\phi)$ of theorem $t$ is an internal node of tree $\mathbf{s}$ and is the root of a subtree; we train the relevance network to imitate this step by selecting this subtree among a large set of candidates.

For a variable $f$ that appears in the assertion but not the hypotheses of $t$, the substitution network is trained to produce its human-written substitute expression $\phi(f)$.

In the case of only human-written theorems but not their proofs, we can no longer perform imitation learning. We instead use reinforcement learning. The objective is to learn actions to maximize the similarity between the generated theorems and human-written theorems. We propose two reward functions to evaluate a generated theorem and update the two deep networks toward the higher rewards via the Reinforce algorithm Williams (1992).

The first reward function is the cross-entropy of a generated theorem given by a language model trained from the human-written theorems. The generator from this reward is called *MetaGen-RL-LM*.

The second reward function is given by an adversarial loss similar to GAN (Goodfellow et al., 2014)—a binary classifier to distinguish the human-written theorems from the generated ones. It is pretrained to separate human-written theorems from the theorems generated by *MetaGen-Rand*, and then updated on-the-fly to separate human-written theorems from the theorems generated by the current generator. The generator is updated to minimize the adversarial loss. We call this generator *MetaGen-RL-Adv*.

More details about the deep networks of the generator are presented in the supplementary material.

## 4.2 Prover

We use Holophrasm (Whalen, 2016) as our theorem prover and augment its training with synthetic data. Given a proof task, Holophrasm conducts backward reasoning to prove the target theorem as described in the supplementary material. For completeness we briefly summarize how Holophrasm works and refer the reader to Whalen (2016) and the supplementary material for more details.

Holophrasm uses Monte Carlo Tree Search (MCTS) to explore multiple branches of actions to find a proof tree. It involves three learnable deep networks: a *payoff network* to determine which branch is

Table 1: Performance of the relevance network of the prover on validation data of `iset.mm` (top two rows) and `set.mm` (starting from the third row).

| Human proofs | Synthetic proofs | Generator | Model | Top-1 | Top-5 | Top-20 | MRR |
|---|---|---|---|---|---|---|---|
| 7123 (ISET) | 0 | - | RELEVANCE | 43.27 | 69.57 | 89.68 | 0.5535 |
| 7123 (ISET) | 1M | *MetaGen-IL* | RELEVANCE | 45.10 | 71.00 | 89.46 | 0.5699 |
| 0 | 0 | - | TF-IDF | 14.28 | 21.13 | 32.55 | 0.1877 |
| 0 | 0 | - | RELEVANCE | 0.96 | 5.33 | 15.67 | 0.0445 |
| 0 | 300K | *MetaGen-Rand* | RELEVANCE | 24.22 | 37.27 | 49.92 | 0.3093 |
| 0 | 300K | *MetaGen-RL-LM* | RELEVANCE | 24.74 | 37.66 | 54.22 | 0.3182 |
| 0 | 300K | *MetaGen-RL-Adv* | RELEVANCE | 25.07 | 39.33 | 50.23 | 0.3242 |
| 2179 (10%) | 0 | - | RELEVANCE | 41.24 | 67.56 | 86.84 | 0.5356 |
| 2179 (10%) | 1M | *MetaGen-Rand* | RELEVANCE | 45.44 | 70.13 | 88.33 | 0.5692 |
| 2179 (10%) | 1M | *MetaGen-IL* | RELEVANCE | 46.10 | 71.12 | 89.38 | 0.5772 |
| 4358 (20%) | 0 | - | RELEVANCE | 47.02 | 72.45 | 89.48 | 0.5870 |
| 21786 (100%) | 0 | - | RELEVANCE | 51.52 | 78.56 | 93.41 | 0.6367 |
| 21786 (100%) | 10M | *MetaGen-Rand* | RELEVANCE | 52.08 | 77.76 | 92.83 | 0.6375 |
| 21786 (100%) | 10M | *MetaGen-IL* | RELEVANCE | 53.20 | 78.73 | 93.13 | 0.6474 |

more promising, a *relevance network* to pick a background theorem to construct a proof step, and a *substitution network*[2] to generate substitutions.

### 4.3 Applicability to other formal systems

As is standard in related work Loos et al. (2017); Irving et al. (2016); Kaliszyk et al. (2018); Yang & Deng (2019), we instantiate and validate our approach on a single formal system, but our approach is applicable to other formal systems such as HOL Light, Coq and Isabelle.

Our approach can be applied to a new system under the following conditions: (1) the search heuristics of the theorem prover can be trained by imitating ground truth proofs; (2) the proof of a theorem is a tree of intermediate goals, and a proof steps demonstrate the entailment of a goal by its children; (3) an intermediate goal in the proof is equivalent to a legal theorem. These conditions are satisfied by the formal systems mentioned above.

To adapt our approach to a new system, the main effort is to rewrite the procedure of sampling proof steps, by replacing substitution with inference rules of the new system. HOL Light, Coq and Isabelle only provide tactics as inference rules to decompose a goal into subgoals for backward reasoning. However, to generate new theorems, we need to execute the corresponding reverse tactics, which are unavailable in their ML environments. We leave the experiments on these systems as future work.

## 5 Experiments

**Dataset** We experiment on two Metamath knowledge bases: `iset.mm` and `set.mm`. `iset.mm` formalizes intuitionistic logic and contains 463 axioms and 8916 theorems, which give rise to 8916 corresponding proof tasks. These proof tasks are divided into 7123 training tasks, 890 validation tasks and 903 test tasks. We use the same version of `set.mm` as Whalen (2016). It formalizes the ZFC set theory and contains 1099 axioms and 27218 theorems, which give rise to 27218 corresponding proof tasks. These proof tasks are divided into 21786 training tasks, 2712 validation tasks and 2720 test tasks.

**Training protocol** On `set.mm`, we control for the number of human proofs provided during training. Specifically, we compare our approach to baselines while including either 0%, 10%, or 100% of the human proofs. We also report the baseline with 20% human proofs for comparison.

Table 2: Performance of the substitution network of the prover on validation data of `iset.mm` (top two rows) and `set.mm` (starting from the third row).

| Human proofs | Synthetic proofs | Generator | Model | Prob | Accuracy |
|---|---|---|---|---|---|
| 7123 (ISET) | 0 | - | SUBSTITUTION | 0.1723 | 49.45 |
| 7123 (ISET) | 1M | *MetaGen-IL* | SUBSTITUTION | 0.2554 | 57.81 |
| 0 | 0 | - | LAUGUAGE MODEL | 0.0032 | 9.06 |
| 0 | 0 | - | SUBSTITUTION | 0.0008 | 0.01 |
| 0 | 300K | *MetaGen-Rand* | SUBSTITUTION | 0.0103 | 29.68 |
| 0 | 300K | *MetaGen-RL-LM* | SUBSTITUTION | 0.0181 | 24.33 |
| 0 | 300K | *MetaGen-RL-Adv* | SUBSTITUTION | 0.0186 | 31.38 |
| 2179 (10%) | 0 | - | SUBSTITUTION | 0.2738 | 58.91 |
| 2179 (10%) | 1M | *MetaGen-Rand* | SUBSTITUTION | 0.3203 | 61.78 |
| 2179 (10%) | 1M | *MetaGen-IL* | SUBSTITUTION | 0.3710 | 66.56 |
| 4358 (20%) | 0 | - | SUBSTITUTION | 0.3765 | 67.07 |
| 21786 (100%) | 0 | - | SUBSTITUTION | 0.6142 | 81.57 |
| 21786 (100%) | 10M | *MetaGen-Rand* | SUBSTITUTION | 0.6439 | 81.85 |
| 21786 (100%) | 10M | *MetaGen-IL* | SUBSTITUTION | 0.6847 | 83.90 |

Table 3: Number of theorems proved on test data of `iset.mm` (top two rows) and `set.mm` (starting from the third row). †: without removing the trivial proof steps from the training data of the relevance network.

| Human proofs | Synthetic proofs | Generator | Prover | Test proofs found |
|---|---|---|---|---|
| 7123 (ISET) | 0 | - | HOLOPHRASM | 378 |
| 7123 (ISET) | 1M | *MetaGen-IL* | HOLOPHRASM | 398 |
| 0 | 0 | - | TF-IDF & LM | 312 |
| 0 | 0 | - | HOLOPHRASM | 219 |
| 0 | 300K | *MetaGen-Rand* | HOLOPHRASM | 346 |
| 0 | 300K | *MetaGen-RL-LM* | HOLOPHRASM | 351 |
| 0 | 300K | *MetaGen-RL-Adv* | HOLOPHRASM | 357 |
| 2179 (10%) | 0 | - | HOLOPHRASM | 454 |
| 2179 (10%) | 1M | *MetaGen-Rand* | HOLOPHRASM | 457 |
| 2179 (10%) | 1M | *MetaGen-IL* | HOLOPHRASM | 472 |
| 4358 (20%) | 0 | - | HOLOPHRASM | 476 |
| 21786 (100%) | 0 | - | HOLOPHRASM('16) | 388 |
| 21786 (100%) | 0 | - | HOLOPHRASM | 557 |
| 21786 (100%) | 10M | *MetaGen-Rand* | HOLOPHRASM | 565 |
| 21786 (100%) | 10M | *MetaGen-IL* | HOLOPHRASM$^{\dagger}$ | 574 |
| 21786 (100%) | 10M | *MetaGen-IL* | HOLOPHRASM | 600 |

**Implementation details** We train the generator on the training set and use the trained generator to generate synthetic theorems and proofs. The prover is trained on both training and synthetic proofs.

On `iset.mm`, we generate 1M unique synthetic theorems. On `set.mm`, we generate 300K unique theorems for the setting of 0% of human proofs (after discarding any duplicates) and 1M unique theorems for 10% of the human training proofs. We generate 10M theorems for the setting of 100% of human proofs, by generating 1M unique theorems a time (maximum allowed by memory limit) and repeating 10 times.

During the training of the relevance network of the prover, we filter out the "trivial" proof steps. A goal is trivial if it is reachable from the assertion of a background theorem $b$ and $b$ has no hypotheses, because this goal can be decomposed by $b$ without generating any new subgoals. By removing the training proof steps that have trivial goals when we train the relevance network, the performance of the prover is improved as shown in Tab. 3.

Please refer to the supplementary material for more details about the implementation and baselines.

### 5.1 Results

To validate the effectiveness of our theorem generator, we evaluate provers trained on the synthetic data and compare them against various baselines.

**Relevance network of prover** We evaluate how synthetic data can improve the relevance network of Holophrasm. The relevance network assigns a score to each candidate background theorem. We use two metrics: (1) top-k accuracy defined as the percentage of times a groundtruth background theorems is ranked in the top k and (2) mean reciprocal rank (MRR) of every groundtruth background theorem among all candidates of its corresponding proof step. Both of them are the higher the better.

We evaluate the relevance network combined with different generators. We also evaluate with tf-idf similarity between sequences of tokens. In Tab. 1, we see that synthetic data brings significant improvement in all settings and the best performance is achieved with our trained generators.

**Substitution network of prover** We evaluate how synthetic data can improve the substitution network of Holophrasm. The substitution network predicts the probability of each token at each position under teacher forcing. We use two metrics: (1) accuracy, defined as the percentage of times the tokens in the groundtruth substitutions have the highest probabilities and (2) the average probability to generate the groundtruth substitutions normalized by its length. Tab. 2 reports the results, including the result of a language model. In all settings, synthetic data brings significant improvement. The best performance is achieved with our trained generators.

**Prover** To evaluate the prover as a whole, we follow the same protocol of Whalen (2016) (more details in the supplementary material) and report the number of theorems proved. We compare with the original Holophrasm prover proposed by Whalen (2016) trained by imitation learning on human-written proofs only. With zero human-written proofs for prover training, we also evaluate TF-IDF & LM, an ablated version of Holophrasm that needs no training proofs—we remove the relevance network and instead pick a background theorem using tf-idf similarity; we replace the substitution network with a language model of theorem statements.

As shown in Tab. 3, the performance of the prover shares the same pattern as the relevance and substitution network. On both `iset.mm` and `set.mm`, the provers trained on synthetic data consistently prove more theorems than the provers trained on human proofs only. On `set.mm`, with 10% human proofs, the use of synthetic proofs almost achieve the same effect by doubling the number of human proofs (472 vs 476 proved theorems). The provers trained with learnable generators perform better than the provers trained with *MetaGen-Rand*.

Our GPU re-implementation of Holophrasm finds 557 proofs trained on 100% of human proofs, more than the number reported in Whalen (2016). We believe this is due to the fact that our prover runs faster on GPUs.

By removing the trivial proof steps from the training data of the relevance network of the prover, the number of proved theorems on the test set increases from 574 to 600.

Polu & Sutskever (2020) demonstrate significant improvement on theorem proving of the `set.mm` benchmark by using very large Transformer (Vaswani et al., 2017) models. Their model can prove 29.22% of test theorems (our percentage is 22.06%). We note a couple potential differences in experimental setup, which may make our results not directly comparable. They appear to use a different version of the `set.mm` knowledge base which has about 38k proofs (ours has 27218 proofs); their evaluation protocol may be different (our prover has a time limit of 5 minutes for each run while their time limit is not mentioned).

Please refer to the supplement material for the examples of synthetic theorems.

## 6 Conclusion

We have proposed a neural generator that automatically synthesizes theorems and proofs for the purpose of training a theorem prover. Experiments on real-world tasks have demonstrated that synthetic data from our approach improves the theorem prover and advances the state of the art of automated theorem proving in Metamath.

**Acknowledgements** This work is partially supported by the National Science Foundation under Grant IIS-1903222 and the Office of Naval Research under Grant N00014-20-1-2634.

## Broader Impact

Our work addresses automated theorem proving. A successful automated theorem prover can help us write programs that are provably correct, which is essential to safety-critical applications, such as software for autonomous driving. On the other hand, since the correctness of the found proofs and synthesized programs relies on the correctness of the underlying theorem prover, bugs in the prover can lead to catastrophic failure.

## Footnotes

[1] Its largest knowledge base, `set.mm` ranks 3rd in the "Formalizing 100 Theorems" challenge (Wiedijk, 2019).

[2]called the generation network in Whalen (2016) but renamed here to avoid confusion with the generator.

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
