[Supplementary Material]

# Learning to Prove Theorems by Learning to Generate Theorems Supplementary Material

## A. Task setup

We use the standard theorem proving setup in prior work Irving et al. (2016); Bansal et al. (2019a); Whalen (2016). Suppose we have a sequence of theorems $(t_1, t_2, ..., t_n)$, where each theorem appear at the order it is proved by mathematicians. For each theorem $t_i$, we construct a proof task that proving $t_i$ (as the target theorem) using all its preceding theorems $(t_1, ..., t_{i-1})$ (as the background theorems), such that the prover has the same set of known facts as mathematicians to prove $t_i$. Then we randomly split the five proof tasks into three sets for training, validation and testing.

It is important to note that a theorem can serve both as a target theorem in the test set and as a background theorem in the training set. This is a standard setup and is not "training on the test set"—a background theorem is used as a known fact in a training proof task and only its statement is provided, not its proof; seeing the statement of a background theorem during training does not tell us how to prove it during testing.

## B. Checking reachability between expressions

For an expression $e$, let $r_e$ be the root node of the parse tree of $e$. Each node in the parse tree represents either a generating axiom (if internal node) or a token (if leaf node). We check if expression $b$ can reach expression $a$ by comparing their parse trees $r_a$ and $r_b$ through the following procedure:

1. Initialize the substitution $\phi$ as empty.

2. Compare the two root nodes.

   - If root node $r_b$ represents a variable $f$, do the following:
     - If the substitute expression $\phi(f)$ is not determined, let $\phi(f) \leftarrow r_a$. Return *True* (i.e. reachable).
     - If $\phi(f) = r_a$, return *True* (i.e. reachable) because we can replace $f$ with $r_a$.
     - Otherwise return *False* (unreachable), because $r_a$ conflicts with the current substitution $\phi$.
   - If the two root nodes represent the same generating aixom or constant, repeat Step 2 to check if each child of $r_a$ is reachable from the corresponding child of $r_b$.
     - If every child of $r_a$ is reachable from the corresponding child of $r_b$, return *True*.
     - Otherwise return *False*.
   - Otherwise return *False*, because the two root nodes have different values and they can not be matched.

This procedure is summarized in Algorithm 1.

---
**Algorithm 1** Function `Reachable`$(n_a, n_b, \phi)$
---
    **Input:** node $n_a$, node $n_b$, substitution $\phi$
    **Output:** *True* if $n_b$ could reach $n_a$, otherwise *False*
    **if** $n_b$ represents a variable $f$ **then**
        **if** $f$ in $\phi$ **then**
            **if** $\phi(f) = n_a$ **then**
                **return** *True* {Consistent with the current substitution}
            **else**
                **return** *False* {Conflict with a preceding branch}
            **end if**
        **else**
            $\phi(f) = n_a$ {Variable $f$ should be replaced by $n_a$}
            **return** *True*
        **end if**
    **else**
        **if** $n_a$ and $n_b$ represent the same generating axiom or constant **then**
            **for** $i = 1$ **to** $\text{len}(c_{n_a})$ **do**
                {$c_n$ is the list of children of node $n$}
                **if** `Reachable`$(c_{n_a}[i], c_{n_b}[i], \phi) = \text{false}$ **then**
                    {A pair of child nodes doesn't match}
                    **return** *False*
                **end if**
            **end for**
            {Every child of $n_b$ could reach a child of $n_a$}
            **return** *True*
        **else**
            **return** *False* {Two nodes have different values}
        **end if**
    **end if**
---

## C. Pseudo-code for MetaGen

Algorithm 3 summarizes the procedure to construct a proof step and the set $S$ of existing proof trees.
Algorithm 4 summarizes the complete procedure of *MetaGen*.

## D. Meaningless theorems

Tree "grafting" can potentially introduce meaningless theorems by combining conflicting hypotheses. For example, suppose the shallow tree proves that $x^2 = 1$ and $x > 0$ entail $x = 1$, we can replace the leaf node $x > 0$ with a subtree proving $x = 5$ entails $x > 0$, which leads to a new tree proving that $x = 5$ and $x^2 = 1$ entail $x = 1$, which is meaningless. Unfortunately, there does not appear to be an easy way to avoid meaningless theorems resulting from tree grafting, because this would require checking the consistency of an arbitrary set of expressions, which can be as hard as general theorem proving. Despite this limitation, however, we still perform tree grafting because a lot of interesting mathematics do result from nontrivial combination of hypotheses.

## E. Holophrasm

In this section we provide more background on the Holophrasm prover Whalen (2016). we refer the reader to Whalen (2016) for more details.

**Backward Reasoning** To construct a proof tree of a target theorem, a straightforward strategy is to search backwards. We start with a single root node—the assertion of the new theorem—and pick a proof step that establishes the entailment of the root node. We expand the tree by adding the preconditions of this proof step as children of the root node. We repeatedly expand the tree by adding children to leaf nodes, until each leaf node is either empty or a hypothesis of the target theorem. This construction process can be understood as recursive goal decomposition: the assertion of the target

---
**Algorithm 2** Initializing existing proof trees
---
    **Input:** existing theorems $E$, existing proofs $P$
    **Output:** existing proof trees $G$
    $G \leftarrow \emptyset$
    **for** theorem $t$ **in** $E$ **do**
        **for** hypothesis $h$ **in** $\mathbf{h}_t$ **do**
            Add $h$ to $G$
        **end for**
        Add $t$ to $G$ as a one-step proof tree
    **end for**
    **for** proof tree $p$ **in** $P$ **do**
        **for** node $e$ **in** $p$ **do**
            $g \leftarrow$ the largest subtree of $p$ rooted at $e$.
            Add $g$ to $G$
        **end for**
    **end for**
---

---
**Algorithm 3** Constructing a proof step
---
    **Input:** existing proof trees $G$, invocable theorems $I$
    **Output:** proof step $(t, \phi)$, proof trees $S$
    Sample an invocable theorem $t \in I$
    $\phi, S \leftarrow \emptyset, \emptyset$
    **for** hypothesis $h$ **in** $\mathbf{h}_t$ **do**
        $C \leftarrow \{\, g \mid g \in G \wedge \mathtt{Reachable}(h, r_g, \phi) \,\}$
        $\{r_g$ is the root node of proof tree $g$. $C$ is the set of compatible existing proof trees$\}$
        Sample a proof tree $g \in C$ using softmax of the relevance network scores
        $\phi' \leftarrow$ the substitution that transforms $h$ to $r_g$
        Add $\phi', g$ to $\phi, S$
    **end for**
    **for** variable $f$ **in** $b$ **do**
        **if** $f$ **not in** $\phi$ **then**
            Generate an expression $e$ using the substitution network
            $\phi(f) \leftarrow e$
        **end if**
    **end for**
---

---
**Algorithm 4** MetaGen
---
    **Input:** existing theorems $E$, existing proofs $P$, int $N$
    **Output:** generated theorems
    Initialize existing proof trees $G$ from $E$ and $P$
    **repeat**
        Construct a proof step $(t, \phi)$ with proof trees $S$
        $g \leftarrow$ the one-step proof tree of $(t, \phi)$
        **for** hypothesis $h$ **in** $\mathbf{h}_t$ **do**
            $\{h(\phi)$ is a leaf node of the one-step proof tree $g\}$
            Find $s \in S$ such that $r_s = h(\phi)$
            Replace $h(\phi)$ with $s$ in $g$ {tree grafting}
        **end for**
        Add the new tree $g$ to $G$
    **until** $G$ reaches the expected volume $N$
---

theorem is the original goal; by picking a proof step we decompose the original goal into subgoals, which are the preconditions of the proof step; then for each subgoal we repeat this process until all subgoals are resolved.

Obviously, each time we expand the tree, we may have multiple choices of proof steps and most of them will lead to dead ends. We thus need to explore multiple alternatives, which gives rise to

Table 1: Training details of the relevance network and the substitution network of the prover.

| Network | Human proofs | Ratio of synthetic proofs steps per batch | Training epochs | Initial learning rate | Epoch to halve learning rate |
|---|---|---|---|---|---|
| RELEVANCE | 0% | 100% | 5 | $10^{-4}$ | - |
| SUBSTITUTION | 0% | 100% | 5 | $5 \times 10^{-4}$ | - |
| RELEVANCE | 10% | 70% | 20 | $10^{-4}$ | [8, 12, 16] |
| SUBSTITUTION | 10% | 70% | 60 | $5 \times 10^{-4}$ | [15, 30, 45] |
| RELEVANCE | 100% | 50% | 16 | $10^{-4}$ | [5, 12, 14] |
| SUBSTITUTION | 100% | 50% | 24 | $5 \times 10^{-4}$ | [10, 15, 20] |

a search process where we need to keep track of what paths have been explored and decide which paths to explore further.

**Proof search** Backward reasoning in Holophrasm Whalen (2016) is implemented with a proof search tree, which keeps track of the exploration of multiple branches of actions to search for a complete proof tree. A proof search tree has two kinds of nodes, expressions and proof steps. An expression node has multiple proof steps as children and each proof step establishes the entailment of this expression by the preconditions. A proof step node has its preconditions as children. A expression is labeled solved if it is a hypothesis of the target theorem or any proof step in its children is solved. A proof step is labeled solved if it has no precondition or all of its preconditions are solved. A complete proof is found if the root node, which is the assertion of the target theorem, is solved.

Holophrasm maintains a payoff of each node in the proof search tree and uses Monte Carlo Tree Search (MCTS) to extend the proof search tree. The prover runs in iterations. In each iteration, it travels down from the root node. After visiting an expression, it either creates a new proof step as a new child or visits its best-performing child according to the UCB (Kocsis & Szepesvári, 2006) algorithm. After visiting a proof step, it travels to its worst-performing child with the lowest payoff. When an expression node is created, it is assigned an initial payoff and has no children. When a proof step node is created, its preconditions are also created as its children and the payoff of this proof step is the lowest payoff among its children. A pass continues until a new proof step is created.

The main heuristics of the prover are how to construct a proof step and what is the initial payoff of an expression. Similar to the generator, the prover constructs a proof step by using a relevance network to pick a background theorem, and a substitution network to generate a substitution for the selected background theorem. The initial payoff of an expression is calculated by a payoff network.

**Relevance network of Holophrasm** The relevance network of the prover is a deep network trained to pick a background theorem $b$ to establish the entailment of an expression $e$, for the purpose of proving a target theorem $t$. It takes as input two sequences of symbols. One sequence represents the assertion and hypotheses of $b$. Another one represents $e$ and the hypotheses of $t$. Two GRU encoders convert each sequence to an embedding vector, followed by a bilinear layer to output a score from two embeddings. The background theorem with the highest score is selected to construct the next proof step. The relevance network is trained to pick the background theorem that is used in the groundtruth proof step.

**Substitution network of Holophrasm** The substitution network generates the substitution for a target variable of a background theorem $b$ for the purpose of proving a target theorem $t$. It is a sequence-to-sequence model with an encoder-decoder GRU network. It takes as input a sequence of symbols that represents the hypotheses of $t$ and the hypotheses of $b$. The target variable is replaced by a special token. It is trained to generate the substitutions of groundtruth proof steps under teacher forcing. When it is called by the prover, it generates multiple substitution candidates for each target variable via beam search.

**Payoff network of Holophrasm** The payoff network calculates the payoff of an expression as the probability of this expression being used in the proof tree of a target theorem. It consists of a GRU network followed by two linear layers and the sigmoid, and takes as input a sequence of symbols that represents the expression to be evaluated and the hypotheses of the target theorem.

The payoff network is trained as a binary classifier to distinguish the expressions in groundtruth proof trees (called positive expressions) from other expressions. Since the payoff network is used to evaluate an expression added to the proof search tree, which is a precondition of a newly generated proof step, the training examples of the payoff network are generated in a similar way. For each positive expression, proof steps that establish the entailment of this expression are constructed by using the pretrained relevance and substitution network. The positive expressions from the preconditions of these proof steps are filtered out and the payoff network is trained to distinguish the positive expressions from the rest of preconditions.

## F. Additional Implementation details

We implement MetaGen and Holophrasm with the same network architectures as used by Whalen (2016). For all of our networks in the generator and the prover, we use bidirectional GRUs to encode input sequences, and use the Adam (Kingma & Ba, 2014) optimizer to update parameters. The batch size is 100 unless otherwise noted.

**Task setup** It is important to note that a theorem can serve both as a target theorem in the test set and as a background theorem in the training set. This is a standard setup and is not "training on the test set"—a background theorem is used as a known fact in a training proof task and only its statement is provided, not its proof; seeing the statement of a background theorem during training does not tell us how to prove it during testing.

**Input representation of the relevance and substitution network** Here we provide more details on the input representation of the relevance and substitution network, which take sequences as input. We use the same form of input representations as used by Whalen (2016).

To represent an expression in a sequential form, one option is to use its "surface form". For example, "(1+1)=2" is simply given as such. Another option is to serialize its parse tree. The parse tree of "(1+1)=2" has two generating axioms. The first axiom is the root node of its parse tree and generates an expression in the form of "A=B". The second axiom is the left child of the root node and generates an expression in the form of "(C+D)" and this expression is used to substitute the variable A in the first axiom. The right child of the first axiom is the token "2". Both of the left child and the right child of the second axiom are the token "1". Then we can represent "(1+1)=2" as a sequence of symbols $(t_=, t_+, 1, 1, 2)$, where each symbol is a node in the parse tree and $t_=$ and $t_+$ represent two generating axioms. This new sequence is obtained by traversing the parse tree in pre-order. Following Whalen (2016), we use the second option to represent expressions as input to our network.

Following Whalen (2016), we also make use of the graph structure of the parse tree. Each node in the input sequence is converted to a feature vector by a learnable embedding layer. Then the feature of this node is concatenated with another four-dimension vector describing the depth of the node, the degree of the node, the degree of its parent, and its position into the children of its parent. The concatenated vector is fed into the GRU encoder of the relevance and substitution network.

Multiple expressions are represented by their concatenation.

### F.1. Generator

**Configuration of GRUs** All of the GRUs in the generator have two layers and 128-dimensional hidden units.

**Training relevance network of *MetaGen-IL*** The relevance network of *MetaGen-IL* is updated to minimize the cross-entropy loss. Each training sample has one groundtruth proof tree and 10 negative candidates that are randomly sampled from compatible proof trees. It is trained for 60 epochs. The learning rate is set to $10^{-4}$ initially and halved after 30, 40 and 50 epochs.

**Training substitution network of *MetaGen-IL*** The substitution network of *MetaGen-IL* is trained for 40 epochs. The learning rate is set to $5 \times 10^{-4}$ initially and halved after 20, 26 and 32 epochs.

**Training of *MetaGen-RL*** To train *MetaGen-RL-LM*, we learn the language model of human-written theorems by utilizing a one-layer GRU with 64-dimensional hidden units. It is trained for 200 epochs. The learning rate is set to $5 \times 10^{-4}$ initially and halved after 80, 120 and 160 epochs.

Table 2: Examples of synthetic theorems from *MetaGen-IL* trained on all human proofs.

| Hypothesis | Assertion | Comment |
|---|---|---|
| $\emptyset$ | $(3 \times 1) + (1 + 0) = 1 + 3$ | SIMPLE ARITHMETIC. |
| $\emptyset$ | $(\log e) \times A = A$ | $e = 2.71828...$ |
| $A \in \mathbb{C}$ $B \in \mathbb{C}$ | $\sin(A + B) = (\exp(\mathbf{i} \times (A + B))$ $- \exp(-\mathbf{i} \times (A + B)) \div (2 \times \mathbf{i})$ | $\mathbb{C}$: COMPLEX NUMBER SET. $\mathbf{i} = \sqrt{-1}$. |
| $\emptyset$ | $G \in \mathbb{R} \wedge E \in \mathbb{R} \to \sin(\frac{G+E}{2} + 1) \in \mathbb{R}$ | $\mathbb{R}$: REAL NUMBER SET. |
| $\phi \to F\colon X \leftrightarrow Y$ | $\phi \to \mathrm{RAN}(F) \subseteq Y$ | F: BIJECTION FROM X TO Y. $\mathrm{RAN}(F)$: RANGE OF $F$. |
| $N = \{x \in \mathbb{Z} \mid M \leq x\}$ | $\phi \wedge K \in N \to$ $M \in \{x \in \mathbb{Z} \mid M \leq x \wedge x \leq K\}$ | $\mathbb{Z}$: INTEGER SET |
| $r = q \times 2 \times y \mod p$ $s = q \times 2 \times x \mod p$ | $x = y \to F(r \times y) = F(s \times x)$ | MOD: MODULO OPERATION |

To train *MetaGen-RL-Adv*, we train a binary classifier using the same architecture as the payoff network of Holophrasm, which contains a two-layer GRU with 128-dimensional hidden units and two subsequent linear layers. It is pretrained to distinguish human-written theorems from 300K synthetic theorems generated by *MetaGen-Rand*. Then it is updated on-the-fly to distinguish human-written theorems from the synthetic theorems generated in the most recent 20 episodes.

For both *MetaGen-RL-LM* and *MetaGen-RL-Adv*, we train the generator for 700 episodes with the learning rate fixed to $10^{-4}$. We deploy 10 parallel threads to synthesize new theorems by utilizing the current generator. Each thread generates 50 theorems in one episode and synchronizes the set $G$ of existing proof trees with other threads for every 20 episodes. We clip policy gradients whose norm is larger 5.

### F.2. Prover

**Configuration of GRUs** In the relevance and substitution network of the prover, all GRUs have two layers and 256-dimensional hidden units. We found 256-dimensional GRUs have slightly better performance than the 128-dimensional GRUs that are used by Whalen (2016). The GRU in the payoff network of the prover has two layers and 128-dimensional hidden units.

**Training of the prover** All three networks of the prover are trained by imitation learning. The relevance network and the substitution network are trained on both human-written proofs and synthetic proofs. The payoff network is trained on human-written proofs only.

The relevance network of the prover is trained to minimize the cross-entropy loss. Each training sample contains one groundtruth background theorem and 10 negative candidates that are randomly sampled from all background theorems that can be applied in this step.

Table 1 presents the settings of learning rate schedules and the ratio of synthetic training samples per batch, for the training of the relevance and substitution network of the prover.

In all experiments, the payoff network is trained for 30 epochs. The learning rate is set to $10^{-4}$ initially and halved after 15, 20 and 25 epochs.

**Evaluation protocol** Following the evaluation protocol used by Whalen (2016), the prover attempts to prove each target theorem in the test set three times with the beam search width of the substitution network set to 1, 5, or 20. The prover stops if it has executed 10000 MCTS passes or hit the time limit of 5 minutes.

### F.3. Baseline

Without human-written proofs, we compare our approach with a baseline that needs no training proofs. We remove the relevance network of the prover and pick a background theorem according to the tf-idf similarity between an expression and a background theorem, as proposed by Bansal et al.

(2019b). We replace the substitution network of the prover with a language model trained on the statements of human-written theorems. We use this language model to generate an expression as the substitution of a target variable.

## G. Examples of generated theorems

Some examples of synthetic theorems are presented in the Table 2. Some are trivial (first and fourth), whereas others are fairly interesting—the third theorem involves a non-trivial statement about trigonometric functions and complex numbers.