[Reviews · NeurIPS 2020]

Review 1

Summary and Contributions: In 2016 Whalen announced the first automated theorem prover using deep reinforcement learning techniques. The prover was based on Metamath, a math formalization corpus and a proof checker. The current work extends Whalen's work by augmentation with synthetic data, which are generated also using neural networks (a combination of deterministic methods, GAN and reinforcement learning).

Strengths: The content of this work is relevant to NIPS. Data augmentation is also a natural and proven methodology to improve machine learning performance, though it is also known that with fixed amount of source data the gain from data augmentation is marginal. Based on a reimplementation of Whalen's code so as to adapt to GPU, the authors have shown some improvements with their data augmentation approach as expected.

Weaknesses: There is not that much novelty in the paper. It would be nice if the authors can give more discussion on the limitations of their approach. And I am wondering whether at certain point of augmentation they have witnessed any degradation of performance due to overfitting. The part on Metamath not clearly written. Being not an expert on Metamath myself, I had to go back to Whalen's paper from time to time (unfortunately it is also not very clearly written) and eventually to Metamath's manual. It took me some time to realize that only the fact of tree interpretation of Metamath sentences is essential to understanding this line of works. I also think it's a bit overclaiming by saying that your work is 'orthogonal' to all previous approaches as your idea verification methodology follows largely from previous approaches, especially Whalen, Loos, etc.

Correctness: Some of the results are overstated, as the paper is not that novel, but other than this there are no complaints about the method and correctness.

Clarity: Some parts could be explained better, as discussed above.

Relation to Prior Work: It is ok.

Reproducibility: Yes

Additional Feedback: 57 -> ``MetaGen'' (correct quotes). 71 -> Reference of HOL-Light (Harrison et al.) 92 -> ...self-play, 'with' one agent... 98 -> 'compete' with each other... 184 -> 'optionally' their proof trees... 322 -> 'MetaGen-Rand' (typo) 324 -> ...due to 'the fact that' our prover runs... 334 -> ...'focus more on'... 341 -> and less 'comfortable' / 'confident' with...


Review 2

Summary and Contributions: The paper describes a 7.7% improvement of the Holophrasm system by generating synthetic theorems. The theorems are generated by guided instantiation and special care is taken not to generate instantiations that are trivially true.

Strengths: Generating relevant synthetic theorems is a nice and nontrivial research direction. It is related to tasks such as lemma introduction and theory exploration. 7.7% more proved theorems on the test set is a decent result. The authors have done a decent amount of work.

Weaknesses: On the other hand, much larger improvements have been achieved by iterating learning and reasoning and thus "synthesizing" the body of training proofs. Such proofs are also created "synthetically" in some sense - as a result of diversely parameterized proof attempts on a curriculum of relevant theorems. This goes back at least to MaLARea [10,11,12] and continues in systems like ATPBoost [13]. See also, e.g., [1,2], where the improvements are 40% and 70% also on the internal guidance tasks. Hence the most straightforward iterative approach to producing more proof data seems far from saturated at the moment. Even in the tactical setting, already the first system (TacticToe) [14] introduces the "self-learning" option of (RL-style) training on its own (i.e. "synthesized") proofs (instead or together with the human proofs). Quite a few other arguments in the paper are incorrect or exaggarated. Already the first version of MPTP and the AI/TP experiments based on it [3] allowed and announced straightforward generation of 630000 related proof tasks from Mizar. Further millions of "synthetic" tasks can be created easily by chasing the large derivation graphs of MML and other ITP libraries. This has been to some extent used later e.g. in works such as [4,5], where the benefit of learning from additional proof tasks was also demonstrated. The same holds about theorem transfer [6], hint-based guidance [7], explicit weakening in Octopus [8,9], etc. In this context, the results are not surprising, and perhaps instead a bit disappointing compared to the large improvements obtained e.g. in [1,2,11,12,13] by using the standard learning/proving loop that generates more and more theorem proving data to learn from. Conjectures have also been synthesized in many other ways, some of them neural [15-22]. Another claim that should be (much) softened is that Holophrasm is "state-of-the-art". It may be the only ATP-style experiment done so far for MetaMath, but the (pretty much universal) experience on more standard ITPs is that the performance of hammers (combined with internal guidance of ATPs, etc.) and of learning-guided tactical systems is today 40-60%. The Metamath theorems are not harder than those in MML, Flyspeck and Isabelle, so a 25-30% performance is most likely improvable by very standard methods. There may be various reasons, e.g., the forward proof style, no pruning by unification (compare with the unification/indexing-based pruning in standard ATPs, etc.) used in Holophrasm. All that said, this seems to be one of the more interesting AI/TP papers submitted to NIPS'20, so I am leaning towards acceptance, provided the incorrect claims are fixed in the final version. ++++++++++++++++++++++++ Update after reading the response and other reviews: After reading the feedback and other reviews, I tend to agree with the rest of the reviewers that the score is still below 6 and the paper may need more work. Holophrasm being state-of-art may be indeed an exaggerated claim. There is no response to the "synthesis" presented here being similar to the one in Octopus. The argument that the synthesis differs from graph chasing and conjecturing is not too convincing either. Analogy-based conjecturing often creates weakenings/strengthenings. Graph-chasing of dependencies and premise selection do it too - just by (not) including some assumptions. I am further reminded of another related work [23], which pushes this quite a bit further. [23] Julio César López-Hernández, Konstantin Korovin: An Abstraction-Refinement Framework for Reasoning with Large Theories. IJCAR 2018: 663-679 ++++++++++++++++++++++ [1] Kaliszyk, C., Urban, J., Michalewski, H. & Olsák, M. Reinforcement Learning of Theorem Proving in NeurIPS 2018 (2018), 8836–8847. [2] Jan Jakubuv, Josef Urban: Hammering Mizar by Learning Clause Guidance. ITP 2019: 34:1-34:8 [3] Urban, J. MPTP - Motivation, Implementation, First Experiments. J. Autom. Reasoning 33, 319– 339 (2004). [4] Cezary Kaliszyk, Josef Urban: Learning-assisted theorem proving with millions of lemmas. J. Symb. Comput. 69: 109-128 (2015) [5] Kaliszyk, C., Urban, J. & Vyskocil, J. Lemmatization for Stronger Reasoning in Large Theories in FroCoS 2015 9322 (Springer, 2015), 341–356. [6] Thibault Gauthier, Cezary Kaliszyk: Sharing HOL4 and HOL Light Proof Knowledge. LPAR 2015: 372-386 [7] Robert Veroff: Using Hints to Increase the Effectiveness of an Automated Reasoning Program: Case Studies. J. Autom. Reasoning 16(3): 223-239 (1996) [8] Monty Newborn, Zongyan Wang: Octopus: Combining Learning and Parallel Search. J. Autom. Reasoning 33(2): 171-218 (2004) [9] M. Newborn: Theo and Octopus at the 2006 World Championship for Automated Reasoning Programs. https://www.cs.mcgill.ca/~newborn/TheoOctopusAtCasc06.pdf [10] Urban, J. MaLARea: a Metasystem for Automated Reasoning in Large Theories in CADE-21 Work- shop on Empirically Successful Automated Reasoning in Large Theories 257 (CEUR-WS.org, 2007). [11] Urban, J., Sutcliffe, G., Pudlák, P. & Vyskocil, J. MaLARea SG1- Machine Learner for Automated Reasoning with Semantic Guidance in IJCAR 2008 5195 (Springer, 2008), 441–456. [12] Kaliszyk, C., Urban, J. & Vyskocil, J. Machine Learner for Automated Reasoning 0.4 and 0.5 in PAAR@IJCAR 2014 31 (EasyChair, 2014), 60–66. [13] Piotrowski, B. & Urban, J. ATPBoost: Learning Premise Selection in Binary Setting with ATP Feedback in IJCAR 2018 10900 (Springer, 2018), 566–574. [14] Gauthier, T., Kaliszyk, C. & Urban, J. TacticToe: Learning to Reason with HOL4 Tactics in LPAR 21 46 (EasyChair, 2017), 125–143. [15] Gauthier, T., Kaliszyk, C. & Urban, J. Initial Experiments with Statistical Conjecturing over Large Formal Corpora in (CICM 2016 - Work in Progress Proceedings 1785 (CEUR-WS.org, 2016), 219– 228. [16] Thibault Gauthier. Deep reinforcement learning in HOL4. CoRR, abs/1910.11797, 2019. [17] Karel Chvalovsky, Thibault Gauthier & Josef Urban: First Experiments with Data Driven Conjecturing, AITP'19, 2019. http://aitp-conference.org/2019/abstract/AITP_2019_paper_27.pdf [18] Josef Urban, Jan Jakubuv: First Neural Conjecturing Datasets and Experiments. CoRR abs/2005.14664 (2020) [19] Lenat, D.B.: AM: an artificial intelligence approach to discovery in mathematics as heuristic search. Ph.D thesis, Stanford (1976) [20] Fajtlowicz, S.: On conjectures of Graffiti. Ann. Discrete Math. 72(1–3), 113–118 (1988) [21] Colton, S.: Automated Theory Formation in Pure Mathematics. Distinguished Dissertations. Springer, London (2012). [22] Johansson, M., Rosén, D., Smallbone, N., Claessen, K.: Hipster: integrating theory exploration in a proof assistant. In: Watt, S.M., Davenport, J.H., Sexton, A.P., Sojka, P., Urban, J. (eds.) CICM 2014. LNCS (LNAI), vol. 8543, pp. 108–122. Springer, Cham (2014).

Correctness: See above.

Clarity: See above.

Relation to Prior Work: See above.

Reproducibility: Yes

Additional Feedback:


Review 3

Summary and Contributions: This paper proposes an approach to augment training data of prover. Trained with the generated data, the the number of theorems proved increased from 557 to 600 as in Table 3, which is the state of the art number on this dataset.

Strengths: The strength of the paper has shown that data augmentation is useful in the field of automated theorem proving. As far as I understood, no researches in the field of ML based automated theorem proving has addressed this direction. As a result, the trained prover achieves the state of the art result on a MetaMath dataset.

Weaknesses: The weakness of the paper is the limited experimental evaluations. The evaluation data is set.mm only and the improvement in Table 3 is not large; I'm not convinced that we are just lucky on this dataset or not. The difference of the number of "test proofs found" between the authors Holophrasm and the previous Holophrasm, the number varies a lot depending configurations. Achieving state of the art is not impactful unless controlling computational cost etc.

Correctness: One approach to show the effectiveness of the proposed approach is to experiment on other Metamath theorem sets. The author proposes a constraint as in line 178. However the effectiveness is not evaluated; adding those analysis can better understanding of proposed data generation method.

Clarity: Combined with the supplementary material, the paper is overall well written. However, the detail of the substitution network is not clear. (Line 229 and 230.) Some example would be helpful.

Relation to Prior Work: The author can explicitly claim the difference or the contribution. However, the difference from the work of Whalen (2016) could have been stated more carefully.

Reproducibility: No

Additional Feedback:

[Author Response · NeurIPS 2020]

Table 1: Performance of the relevance and substitution networks of the prover on `iset.mm` validation data.

| Human proofs | Synthetic proofs | Generator | Relevance | | | | Substitution | | Test proofs found (903 in total) |
|---|---|---|---|---|---|---|---|---|---|
| | | | Top-1 | Top-5 | Top-20 | MRR | Prob | Accuracy | |
| 7125 | 0 | - | 43.27 | 69.57 | 89.68 | 0.5535 | 0.1723 | 49.45 | 378 |
| 7125 | 1M | *MetaGen-IL* | 45.10 | 71.00 | 89.46 | 0.5699 | 0.2554 | 57.81 | 398 |

We thank all reviewers for their thoughtful comments. As suggested by R3, we experimented on another Metamath
theorem sets, `iset.mm`, and the results are presented in the table above. Individual questions are addressed below.

**R1–There is not that much novelty in the paper.** Our novelty is that we propose to generate synthetic theorems by
forward reasoning as training data for the theorem prover. While most of prior work uses training data from the proofs
(or their variants) of human-written theorems, we demonstrate that the synthetic proofs, which do not contribute to
proving human-written theorems, can also be effectively used to advance the prover on human-written theorems. This
implies that to train a powerful theorem prover, not only should the prover learn to remember the proofs of given
human-written theorems, but also learn to reason about the entire space of potential theorems defined by the existing
theorems. We believe this is an important direction that worth more exploration in the AI/TP community.

**R1–Give more discussion on the limitations of the proposed approach.** We list three conditions to apply our
approach to other formal systems in Sec. 4.3, which is one aspect of the limitations of our approach. Another limitation
is that we assume a set of human-written theorems as inputs. We expect to explore in the future work the generation of
plausible and meaningful synthetic theorems from the basic definitions and axioms only.
**R1–The part on Metamath not clearly written.** We will clarify the part of Metamath as you suggested.
**R1–It's a bit overclaiming by saying that your work is 'orthogonal' to all previous approaches.** We believe our
main idea is orthogonal to prior work. Our idea is to generate synthetic theorems as training data for the prover, which
can also be applied to prior work of neural theorem proving. But we will tone down our text and further clarify.
**R1–Any degradation of performance due to overfitting?** No. We believe our synthetic theorems do not cause
overfitting for two reasons: (1) synthetic theorems are randomly sampled. (2) every new theorem has a unique proof
that is different from existing proofs and it contains a novel path of deduction that is not covered by existing data.

**To R2:** Thank you for pointing us to a list of related work. We discuss how our work differs from these related works
below. We will add these discussions to our next revision and revise any inaccurate statements.
**R2–Prior work on iterative learning and reasoning.** In iterative learning and reasoning, machine-proofs are
generated for existing human theorems and are used to train the prover. That is, only new proofs are synthesized and
the new proofs are only for existing human theorems, but no new theorems are synthesized. In contrast, our approach
synthesize both new theorems and new proofs. Our synthetic theorems and their proofs could cover a much larger space
of possible derivations than the proofs of existing human theorems.
**R2–Prior work ([3,4,5] from R2) on generating synthetic proof tasks.** [3,4,5] extract synthetic proof tasks from the
proofs of human-written theorems, such as the intermediate steps or their variants. That is, they extract "sub-proofs"
from existing proofs. In contrast, we generate entirely new theorems and new proofs that are not part of any existing
proofs.
**R2–Prior work on conjecturing.** Conjecturing targets on finding meaningful math theorems automatically. Generated
conjectures could be either true or false and their proofs are not required. In contrast, each of our synthetic theorem is
guaranteed to be correct and its proof is automatically available.
**R2–The claim that Holophrasm is "state-of-the-art" should be softened.** Thank you for the suggestion. We will
soften this language.

**R3–Limited experimental evaluations.** We have now experimented on another Metamath theorem sets `iset.mm`
which formalizes 9371 theorems in intuitionistic logic. As shown in Tab. 1, the prover trained with both human-written
data and synthetic data performs better than the Holophrasm baseline, which is consistent with our results on `set.mm`.
**R3–Achieving state of the art is not impactful unless controlling computational cost.** Our claim of SOTA is against
our re-implementation of the baseline in the GPU environment. Thus the computation cost is already controlled for.
We will further clarify our claim and release our code such that the future work could compare with us in the GPU
environment.
**R3–Evaluate the effectiveness of the proposed constraints in Ln. 178** These constraints (that limit what can be
invocable theorems) are proposed to simplify our proposed method and to reduce computational cost. Without them,
training the prover will be much more costly because there will be more invocable theorems. We are not able to
complete this experiment for this rebuttal but will add it to our next revision.
**R3–The detail of the substitution network is not clear.** We will clearify the substitution network by adding examples.

[Meta-Review · NeurIPS 2020]

We have a very thorough review for this paper giving it a rating of 6. The topic seems relevant and important but the scores are borderline. The main issue seems to be properly placing the work in the context of previous work with proper comparisons. There were also some complaints about the clarity of the presentation. The paper would greatly benefit from a rewrite and probably extended experiments taking these reviews into consideration.